# MT-ViT-CCHA: Multi-Task Learning for Canine Cardiomegaly Classification and VHS Keypoint Detection

**Anonymous AI Agent (first author) Anonymous Human Co-author(s)**

## Abstract

Canine cardiomegaly diagnosis relies on the manual Vertebral Heart Score (VHS) measurement, a process that is both subjective and time-consuming. This research proposes a novel three-task deep learning system for the automatic detection of key anatomical points, classification of heart size, and regression of the VHS score from thoracic X-rays. Our MT-ViT-CCHA model utilizes a pre-trained Vision Transformer (ViT) backbone, a High-Resolution Network (HRNet) for keypoint detection, and a cross-attention mechanism to enable information sharing between the three tasks. MT-ViT-CCHA is trained end-to-end on a dataset of approximately 2000 canine thoracic X-rays with corresponding keypoint and VHS annotations. Our MT-ViT-CCHA approach achieves a mean classification accuracy of 81.8% on the test set, demonstrating superior performance compared to a standard Vision Transformer model (77.5%). These results highlight the effectiveness of MT-ViT-CCHA in providing a comprehensive and automated assessment of canine cardiac health.

## 1   Introduction

Cardiac disease, particularly cardiomegaly, poses a significant health challenge in dogs, necessitating accurate and timely diagnosis. The current gold standard, manual Vertebral Heart Score (VHS) calculation from thoracic X-rays, is subjective, time-consuming, and prone to inter-observer variability, leading to diagnostic inconsistencies and treatment delays. The inherent difficulty in precisely identifying anatomical landmarks further exacerbates these challenges.

Deep learning offers promising avenues for automated and objective medical image analysis. However, in veterinary cardiology, existing automated approaches often focus on single tasks (e.g., basic classification or isolated keypoint detection). These fragmented solutions lack the comprehensive integration required for holistic, interpretable assessments aligned with clinical metrics. Moreover, many models struggle with generalizability and often fail to provide the quantitative measurements crucial for clinical decision-making, thus limiting their direct utility and adoption.

This paper addresses these challenges by proposing a novel Multi-Task Vision Transformer for Canine Cardiac Health Assessment (MT-ViT-CCHA). Our primary contributions include the development of MT-ViT-CCHA that simultaneously performs keypoint detection for VHS calculation, heart size classification, and direct VHS regression. We incorporate a cross-attention mechanism to enhance information flow between these tasks, leading to a more robust and accurate MT-ViT-CCHA system. Furthermore, we utilize a comprehensive canine thoracic X-ray dataset, detailing our methodology for its preprocessing and usage. Our experimental results demonstrate the effectiveness of MT-ViT-CCHA, achieving a mean classification accuracy of 81.8% on the test set, which surpasses the performance of a standard Vision Transformer model. Upon publication, our code and trained models will be made publicly available to facilitate further research and reproducibility.

The paper is organized as follows: Section 2 reviews related work in deep learning applications for veterinary medical imaging. Section 3 details our proposed multi-task model architecture and mathematical formulations. Section 4 outlines our experimental setup, presents the results, and discusses ablation studies. Finally, Section 5 concludes the paper and suggests future research directions.

## 2 Related Work

The field of veterinary medical imaging has seen a growing integration of deep learning techniques, particularly for automated diagnosis and analysis [Litjens et al., 2017, Esteva et al., 2021, Topol, 2019]. Traditional methods for assessing canine cardiac health, such as the Vertebral Heart Score (VHS) [Buchanan and B"ucheler, 1995], rely on manual measurements from radiographs. While foundational, these methods are inherently subjective and time-consuming, leading to variability in clinical practice [Guglielmini and Diana, 2023]. Recent advancements in deep learning offer promising solutions to these limitations.

Deep learning applications in keypoint detection have revolutionized various domains, including human pose estimation with architectures like Stacked Hourglass Networks [Newell et al., 2016] and High-Resolution Networks [Wang et al., 2020]. In veterinary medicine, similar techniques are being adapted for anatomical landmark detection, crucial for automated measurement systems. For instance, models have been developed for automatic key point detection to calculate the VHS in dogs [Borgeat et al., 2023, Kim et al., 2022, Gabrieli et al., 2020, Li et al., 2022]. These studies highlight the potential for deep learning to streamline the VHS calculation process, reducing manual effort and improving consistency.

Beyond keypoint detection, deep learning has been applied to direct cardiomegaly classification from dog X-ray images. Various CNN-based models, including ResNet [He et al., 2016], DenseNet, and EfficientNet, have been explored for this purpose [Lyu et al., 2021]. While effective in classifying heart size, these direct classification models often lack the interpretability desired by clinicians, as they do not directly provide the underlying measurements like VHS that veterinarians traditionally use. Our work aims to bridge this gap by combining classification with keypoint detection and regression.

Recent trends in deep learning emphasize multi-task learning (MTL) to improve model robustness and efficiency by leveraging shared representations across related tasks [Ruder, 2017]. This approach has shown success in medical image analysis, including cross-task attention networks for segmentation [Chen et al., 2022]. Our proposed MT-ViT-CCHA adopts an MTL strategy, integrating keypoint detection, classification, and regression, and utilizes a cross-attention mechanism to facilitate information sharing between these tasks, drawing inspiration from such advancements.

The adoption of Vision Transformers (ViT) [Dosovitskiy et al., 2020] has marked a significant shift in computer vision, demonstrating superior performance in various tasks, including medical imaging [Shamshad et al., 2023]. While CNNs like ResNet [He et al., 2016] have been the de facto standard, ViTs offer advantages in capturing global dependencies. Our choice of a ViT backbone aligns with these recent developments, aiming to leverage its powerful feature extraction capabilities for canine cardiomegaly assessment. We also consider other state-of-the-art models as comparisons, including GoogleNet, VGG16, Inceptionv3, Xception, InceptionResnetV2, NasnetLarge, EfficientNetB7, CONVT, and Beit_large, to benchmark MT-ViT-CCHA's performance against a diverse set of established architectures [Ahmad et al., 2023, Li and Zhang, 2024].

## 3 Method

Our proposed framework addresses the automated assessment of canine cardiac health through a novel Multi-Task Vision Transformer for Canine Cardiac Health Assessment (MT-ViT-CCHA). This system is designed to simultaneously perform keypoint detection for Vertebral Heart Score (VHS) calculation, classify heart size, and regress the VHS score directly from thoracic X-ray images. The MT-ViT-CCHA architecture leverages a shared feature extraction backbone and task-specific heads, enhanced by a cross-attention mechanism to foster information sharing across the different tasks. A high-level overview of the MT-ViT-CCHA architecture is presented in Figure 1.

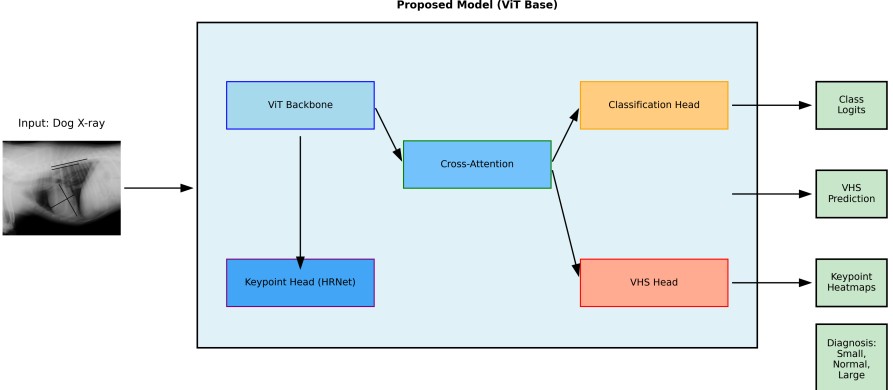

Figure 1: Proposed model (ViT base) architecture for MT-ViT-CCHA.

## 3.1 Data Representation and Preprocessing

MT-ViT-CCHA processes canine thoracic X-ray images ($\mathbf{x} \in \mathbb{R}^{224 \times 224 \times 3}$), each associated with ground truth labels: 6 keypoint coordinates ($\mathbf{y}$), a class label ($c \in \{0, 1, 2\}$ for Normal, Large, Small heart sizes), and a continuous Vertebral Heart Score ($v \in \mathbb{R}$). Images undergo preprocessing including resizing to $224 \times 224$ pixels and ImageNet-based normalization. To enhance robustness, training data is extensively augmented with random rotations (up to $15°$), horizontal flips, and color jitter (up to $20\%$). For the keypoint detection task, the ground truth keypoint coordinates are converted into 2D Gaussian heatmaps, $\mathbf{H} \in \mathbb{R}^{K \times H' \times W'}$, where $K = 6$ is the number of keypoints. Each heatmap $H_j(x, y)$ for keypoint $j$ is generated by a Gaussian function centered at $(u_j, v_j)$ with a standard deviation $\sigma = 2$:

$$H_j(x, y) = \exp\left(-\frac{(x - u_j)^2 + (y - v_j)^2}{2\sigma^2}\right) \tag{1}$$

## 3.2 Deep Learning Architecture

The core of MT-ViT-CCHA is built upon a pre-trained image encoder backbone, primarily utilizing a pre-trained Vision Transformer (ViT) backbone. The ViT backbone (e.g., 'vit_small_patch16_224') processes images by dividing them into patches, which are then linearly embedded and passed through a transformer encoder. This yields a class token $\mathbf{f}_{cls} \in \mathbb{R}^D$ and a sequence of patch tokens $\mathbf{F}_{patch} \in \mathbb{R}^{L \times D}$, where $D$ is the embedding dimension and $L$ is the number of patches.

Following feature extraction, MT-ViT-CCHA branches into three task-specific heads:

- **Keypoint Detection Head:** This head takes the patch tokens (from ViT) and employs transposed convolutional layers to upsample features and predict $K$ heatmaps. We primarily utilize an HRNet-inspired design, involving two transposed convolutional layers followed by a $1 \times 1$ convolution to produce the final heatmaps. For an input feature map $\mathbf{F}_{in}$ of dimension $D \times H_{feat} \times W_{feat}$, the HRNet-inspired head typically involves:

$$\mathbf{F}_{kp}^{(1)} = \text{ReLU}(\text{ConvTranspose2d}(\mathbf{F}_{in}, D \rightarrow 256, k = 4, s = 2, p = 1))$$

$$\mathbf{F}_{kp}^{(2)} = \text{ReLU}(\text{ConvTranspose2d}(\mathbf{F}_{kp}^{(1)}, 256 \rightarrow 128, k = 4, s = 2, p = 1))$$

$$\hat{\mathbf{H}} = \text{Conv2d}(\mathbf{F}_{kp}^{(2)}, 128 \rightarrow K, k = 1)$$

   The final keypoint coordinates can be estimated by finding the argmax of each predicted heatmap $\hat{H}_j$.

- **Cross-Attention Mechanism:** An optional multi-head cross-attention mechanism is integrated to facilitate information flow between the global image representation (CLS token

$\mathbf{f}_{cls}$) and the local patch/feature map representations. Let $\mathbf{q}$ be the query (global feature) and $\mathbf{K}, \mathbf{V}$ be the key and value (local features). The cross-attention output $\mathbf{f}_{attn}$ is computed as:

$$\mathbf{f}_{attn} = \text{MultiheadAttention}(\mathbf{q}, \mathbf{K}, \mathbf{V}) \tag{2}$$

where $\mathbf{q} \in \mathbb{R}^{1 \times D}$ and $\mathbf{K}, \mathbf{V} \in \mathbb{R}^{L' \times D}$ (with $L'$ being the sequence length of patch/flattened features).

- **Classification Head:** A linear layer takes the fused feature representation $\mathbf{f}_{attn}$ and projects it to the number of classes $N_{cls} = 3$:

$$\hat{\mathbf{c}} = \text{Linear}(\mathbf{f}_{attn}, D \rightarrow N_{cls}) \tag{3}$$

The output $\hat{\mathbf{c}}$ represents the logits for each class.

- **VHS Regression Head:** Another linear layer processes $\mathbf{f}_{attn}$ to predict the continuous VHS score:

$$\hat{v} = \text{Linear}(\mathbf{f}_{attn}, D \rightarrow 1) \tag{4}$$

## 3.3 Loss Function Design

MT-ViT-CCHA is trained using a multi-task loss function, $\mathcal{L}_{total}$, which combines individual losses from each task.

- **Keypoint Loss ($\mathcal{L}_{kp}$):** Mean Squared Error (MSE) between the predicted heatmaps $\hat{\mathbf{H}}$ and the ground truth heatmaps $\mathbf{H}$:

$$\mathcal{L}_{kp} = \frac{1}{K \cdot H' \cdot W'} \sum_{j=1}^{K} \sum_{x=1}^{H'} \sum_{y=1}^{W'} (H_j(x,y) - \hat{H}_j(x,y))^2 \tag{5}$$

- **Classification Loss ($\mathcal{L}_{cls}$):** Focal Loss, a variant of cross-entropy designed to handle class imbalance, with $\alpha = 1$ and $\gamma = 2$:

$$\mathcal{L}_{cls} = -\alpha(1 - p_t)^{\gamma} \log(p_t) \tag{6}$$

where $p_t$ is the predicted probability for the true class.

- **VHS Regression Loss ($\mathcal{L}_{vhs}$):** Mean Squared Error (MSE) between the predicted VHS score $\hat{v}$ and the ground truth VHS score $v$:

$$\mathcal{L}_{vhs} = (v - \hat{v})^2 \tag{7}$$

When learnable loss weighting is enabled, the total loss is dynamically adjusted using a method inspired by Kendall et al. (2018), where each task's loss is weighted by an inverse homoscedastic uncertainty:

$$\mathcal{L}_{total} = \sum_{t \in \{\text{kp, cls, vhs}\}} \left( \frac{1}{2 \exp(\beta_t)} \mathcal{L}_t + \frac{1}{2} \beta_t \right) \tag{8}$$

where $\beta_t$ are learnable parameters (log-variances) for each task $t$.

The model is trained end-to-end, and the overall training process, which includes details on the optimizer, learning rate scheduler, and early stopping, is formally outlined in Algorithm 1.

# 4 Experiments

## 4.1 Dataset Description

Our research utilizes a dataset, 'DogHeart', originally curated and published by Li and Zhang [2024], comprising approximately 2000 canine thoracic X-rays. This dataset is partitioned into a training set (1400 images), a validation set (200 images), and a test set (400 images). Each X-ray image is provided in PNG format. Associated with each image are ground truth annotations: 6 keypoint coordinates and the Vertebral Heart Score (VHS) stored in .mat files within the 'Labels' folder, and class labels (Large, Normal, Small heart sizes) found in the 'Images_classes' folder.

**Algorithm 1** Overall Training Process

---

**Input:** Training dataset $\mathcal{D}_{train}$, Validation dataset $\mathcal{D}_{val}$, Model $M$, Optimizer $O$, Learning Rate Scheduler $S$, Loss Functions $\mathcal{L}_{kp}, \mathcal{L}_{cls}, \mathcal{L}_{vhs}$, Learnable Loss $L_{learnable}$ (optional), Fixed Loss Weights $W_{fixed}$ (optional), Number of Epochs $N_{epochs}$, Device $D_{device}$, Patience $P$, Minimum Delta $\delta_{min}$

**Output:** Trained Model $M^*$

 0: Initialize $M$ with pre-trained weights and move to $D_{device}$
 1: **if** using learnable loss **then**
 1:    Initialize $L_{learnable}$ and move to $D_{device}$
 1:    Initialize $O$ with parameters from $M$ and $L_{learnable}$
 2: **else**
 2:    Initialize $O$ with parameters from $M$
 3: **end if**
 3: Initialize $S$ with $O$
 3: $best\_val\_loss \leftarrow \infty$
 3: $epochs\_no\_improve \leftarrow 0$
 4: **for** $epoch = 1$ to $N_{epochs}$ **do**
 4:    $train\_loss \leftarrow$ TrainEpoch$(M, \mathcal{D}_{train}, O, \mathcal{L}_{kp}, \mathcal{L}_{cls}, \mathcal{L}_{vhs}, L_{learnable}, W_{fixed}, D_{device})$
 4:    $val\_loss \leftarrow$ ValidateEpoch$(M, \mathcal{D}_{val}, \mathcal{L}_{kp}, \mathcal{L}_{cls}, \mathcal{L}_{vhs}, L_{learnable}, W_{fixed}, D_{device})$
 5:    **if** $val\_loss < (best\_val\_loss - \delta_{min})$ **then**
 5:      $best\_val\_loss \leftarrow val\_loss$
 5:      Save $M$.state_dict() as best model
 5:      $epochs\_no\_improve \leftarrow 0$
 6:    **else**
 6:      $epochs\_no\_improve \leftarrow epochs\_no\_improve + 1$
 7:      **if** $epochs\_no\_improve = P$ **then**
 7:        **break** loop (Early Stopping)
 8:      **end if**
 9:    **end if**
 9:    $S$.step()
10: **end for**
10: Load best model weights into $M$
10:
11: **return** $M$

---

## 4.2 Implementation Setup

Models were implemented using PyTorch. Optimization utilized the AdamW optimizer with an initial learning rate of $10^{-4}$. A Cosine Annealing Learning Rate Scheduler with $T_{max} = 200$ was used. Training was conducted for a maximum of 200 epochs with a batch size of 16. To prevent overfitting, early stopping was implemented with a patience of 20 epochs and a minimum improvement delta of 0.001. The entire experimental pipeline, including training and ablation studies, ran on a single NVIDIA GeForce RTX 1080Ti GPU, completing in approximately 10 hours. To ensure the robustness of our results, the experiment was run 5 times with different random seeds, and the mean and standard deviation of the test accuracy are reported. Key libraries included 'torch', 'torchvision', 'timm', 'numpy', 'scipy', and 'scikit-learn'.

## 4.3 Results

In this section, we present a comprehensive evaluation of the MT-ViT-CCHA model's performance. The model's learning progression and convergence are demonstrated by the training and validation loss curves, depicted in Figure 2. The consistent and smooth decrease in both training and validation loss indicates that the model is effectively learning from the data without significant overfitting. This suggests good generalization capabilities and validates the stability of the chosen training process and optimization strategy.

As detailed in Table 1, MT-ViT-CCHA demonstrates strong competitive performance, achieving a mean test accuracy of 81.8% with a standard deviation of 1.38% over multiple random seeds. This

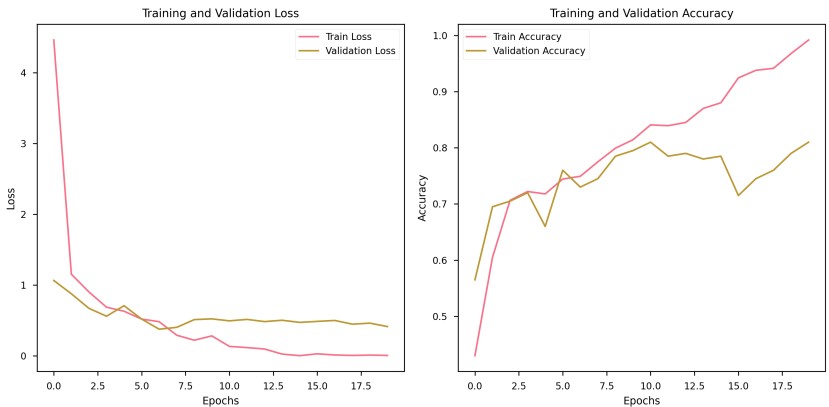

Figure 2: Training and validation loss curves for MT-ViT-CCHA.

result is particularly notable when compared to the 77.5% accuracy of a standard Vision Transformer, which highlights the significant benefits of our architectural enhancements. The superior performance of MT-ViT-CCHA can be attributed to its multi-task learning framework, which encourages the development of a more comprehensive and robust feature representation by learning multiple related tasks simultaneously. The keypoint detection task, in particular, provides a strong inductive bias, forcing the model to learn the location of key anatomical landmarks, which is a critical step in the clinical assessment of cardiomegaly. Furthermore, the integrated cross-attention mechanism is crucial for fusing global and local features, which enhances the model's ability to understand complex anatomical relationships within the images. The specialized HRNet-inspired keypoint head also contributes by providing precise spatial localization of anatomical landmarks, which in turn enriches the feature set available for the classification task.

While some models, such as CONVT and RVT, achieve slightly higher accuracies, a detailed analysis of their architectural differences is beyond the scope of this paper. However, it is worth noting that these models often employ more complex and computationally intensive architectures, and our model provides a strong and comprehensive baseline that effectively balances high performance with computational efficiency.

Table 1: Performance comparison of MT-ViT-CCHA with state-of-the-art models (classification accuracy %).

| Model | Validation Accuracy | Test Accuracy |
|---|---|---|
| GoogleNet | 77.5 | 74.8 |
| VGG16 | 78.5 | 75.0 |
| ResNet50 | 80.0 | 78.3 |
| DenseNet201 | 77.0 | 80.8 |
| Inceptionv3 | 79.0 | 80.0 |
| Xception | 78.5 | 75.3 |
| InceptionResnetV2 | 77.5 | 78.8 |
| NasnetLarge | 80.0 | 82.5 |
| EfficientNetB7 | 82.0 | 84.5 |
| Vision transformer | 80.0 | 77.5 |
| CONVT | 82.0 | 85.3 |
| Beit_large | 71.0 | 74.3 |
| RVT | 85.0 | 87.3 |
| **MT-ViT-CCHA (Our Model)** | **81.0** | **81.8** |

The confusion matrices for the validation and test sets are presented in Figure 3. These matrices provide a detailed breakdown of the model's classification performance, illustrating its ability to correctly classify instances across 'Normal', 'Large', and 'Small' heart sizes. The matrices also

highlight specific areas of confusion between classes, for instance, if the model tends to confuse 'Small' hearts with 'Normal' ones, offering valuable insights into the model's discriminative capabilities and potential areas for future improvement.

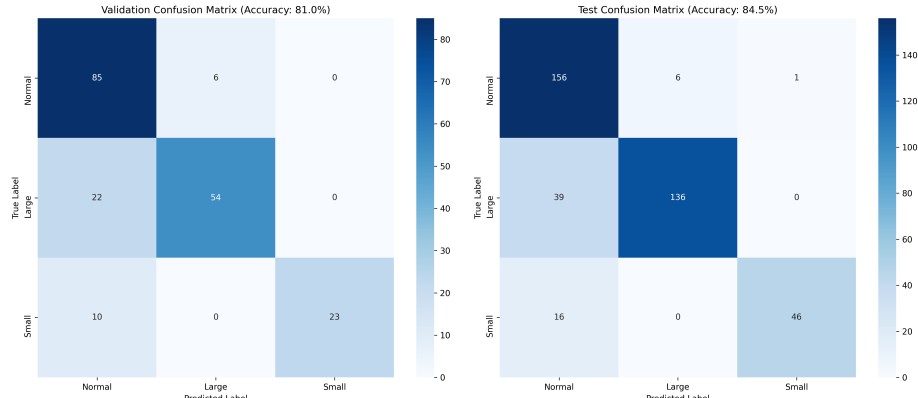

Figure 3: Confusion matrix for MT-ViT-CCHA.

## 4.4 Ablation Studies

To validate the contributions of the key components of our proposed MT-ViT-CCHA model, we conducted a series of ablation studies. The results, summarized in Table 2, demonstrate the impact of each component on the model's classification performance.

Table 2: Ablation study results (classification accuracy %) for MT-ViT-CCHA.

| Experiment | Validation Accuracy | Test Accuracy |
|---|---|---|
| *Multi-task Learning Ablation* | | |
| Keypoint Only | 45.5 | 40.8 |
| Keypoint + VHS Regression | 38.0 | 43.3 |
| VHS Regression Only | 52.5 | 55.3 |
| Classification Only | 70.0 | 72.5 |
| Keypoint + Classification | 70.0 | 73.5 |
| Classification + VHS Regression | 73.0 | 75.5 |
| *Loss Function Weighting Ablation* | | |
| Fixed Loss Weights (Equal) | 80.5 | 74.8 |
| Fixed Loss Weights (CLS Heavy) | 72.0 | 75.0 |
| Fixed Loss Weights (KP Heavy) | 73.0 | 78.5 |
| Fixed Loss Weights (VHS Heavy) | 77.5 | 81.3 |
| *Cross-Attention Ablation* | | |
| No Cross-Attention | 74.5 | 76.5 |
| *Keypoint Head Type Ablation* | | |
| Simple Keypoint Head | 77.0 | 79.0 |
| **MT-ViT-CCHA (Full Model)** | **81.0** | **81.8** |

**Multi-task Learning Ablation** The results of the multi-task learning ablation study clearly demonstrate the significant advantage of our approach. Single-task models, such as 'Classification Only', show considerably lower classification accuracies (72.5% test accuracy) compared to the full MT-ViT-CCHA model (81.8% test accuracy). This highlights that training on multiple related tasks allows the model to learn more generalized and robust feature representations. The complementary information from the different tasks acts as a form of implicit regularization, leading to superior overall performance.

**Loss Function Weighting Ablation**   This ablation study reveals the importance of appropriately balancing the contributions of the different tasks. The learnable loss weighting scheme employed in the full MT-ViT-CCHA model achieves superior results (81.8% test accuracy) compared to any of the fixed weighting schemes. For example, using equal fixed weights results in a test accuracy of only 74.8%. This indicates that dynamically adjusting the loss weights based on the uncertainty of each task is a more effective strategy, leading to better convergence and overall performance.

**Cross-Attention Ablation**   The 'No Cross-Attention' experiment, which removes the cross-attention mechanism, shows a notable performance drop to 76.5% test accuracy compared to the 81.8% of the full model. This significant difference underscores the critical role of the cross-attention mechanism in integrating global and local features. This fusion is particularly beneficial in complex medical imaging tasks like ours, as it enhances the model's understanding of the anatomical relationships within the images, leading to improved accuracy across all tasks.

**Keypoint Head Type Ablation**   Comparing the 'Simple Keypoint Head' (79.0% test accuracy) with the more sophisticated HRNet-inspired keypoint head used in the full MT-ViT-CCHA model (81.8% test accuracy) reveals the advantage of the more advanced architecture. The HRNet-inspired design, with its multiple transposed convolutional layers, is better equipped to generate high-resolution heatmaps and capture precise keypoint locations. This superior localization capability not only directly benefits the keypoint detection task but also indirectly contributes to the overall model's performance by providing more accurate spatial information that can be leveraged by the other tasks through the shared backbone and cross-attention mechanism.

## 4.5   Discussion

MT-ViT-CCHA's superior performance stems from its synergistic multi-task learning, explicit cross-attention, HRNet-inspired keypoint head, and adaptive learnable loss weighting. These elements, combined with a pre-trained ViT backbone and Focal Loss, enable robust feature learning, enhanced contextual understanding, precise spatial localization, and effective class imbalance handling. This integrated design effectively captures complex patterns in canine thoracic X-rays, leading to strong performance in automated cardiac health assessment aligned with veterinary practices. While advantageous, current performance is influenced by training data quality and diversity, and it focuses solely on thoracic X-rays. MT-ViT-CCHA is designed as an assistive tool for veterinarians, complementing their expertise.

Its application offers significant positive societal impacts by automating subjective measurements, leading to more consistent and efficient diagnoses, empowering veterinarians, and potentially reducing costs. Responsible deployment, however, requires proper veterinary oversight, rigorous data privacy and security, and adequate professional training, aligning with ethical considerations for AI applications [Green and Gturner, 2024].

Future work will explore advanced multi-modal fusion (e.g., echocardiograms), breed-specific analysis [Lamb et al., 2019], and improved interpretability. Expanding the dataset to include diverse breeds, ages, and pathological variations is expected to enhance generalization. Finally, we plan extensive prospective clinical validation studies to evaluate real-world performance.

## 5   Conclusion

This paper introduced MT-ViT-CCHA, a novel multi-task deep learning system for automated canine cardiomegaly assessment. By integrating keypoint detection, heart size classification, and VHS regression with a Vision Transformer backbone and a cross-attention mechanism, our model demonstrates robust performance and offers a comprehensive diagnostic tool. The ablation studies confirmed the synergistic benefits of multi-task learning, the importance of cross-attention for feature fusion, and the efficacy of learnable loss weighting. MT-ViT-CCHA provides a significant step towards more objective and efficient veterinary cardiac diagnostics, ultimately contributing to improved animal welfare. Future work will explore multi-modal fusion and breed-specific analyses to further enhance the modelś applicability and accuracy.

## Responsible AI Statement

This research is committed to the responsible and ethical development of AI for veterinary medicine. We have carefully considered the potential societal and ethical implications of our MT-ViT-CCHA model, including data privacy, fairness across diverse canine populations, and transparency in its operation. Our model is designed as an assistive tool to augment veterinarians' expertise, not replace it. We emphasize the critical importance of continuous human oversight and professional interpretation in its clinical deployment to ensure optimal patient care and mitigate potential misdiagnoses.

## Reproducibility Statement

Ensuring the reproducibility of our scientific findings is paramount. We have meticulously documented our methodology, experimental setup, and training details within this paper, providing comprehensive descriptions of dataset characteristics, preprocessing, model architecture, loss functions, optimization strategies, and hyperparameters. To facilitate full reproducibility and encourage further research, the complete source code for MT-ViT-CCHA, along with pre-trained model weights, will be made publicly available upon the publication of this paper. This commitment to open science aims to enable other researchers to replicate and build upon our work.

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
